# AceReason-Nemotron: Advancing Math and Code Reasoning through Reinforcement Learning

Yang Chen [*]       Zhuolin Yang [*]       Zihan Liu       Chankyu Lee       Peng Xu

Mohammad Shoeybi              Bryan Catanzaro              Wei Ping

NVIDIA

## Abstract

Despite recent progress in large-scale reinforcement learning (RL) for reasoning, the training recipe for building high-performing reasoning models remains elusive. Key implementation details of frontier models, such as DeepSeek-R1, including data curation strategies and RL training recipe, are often omitted. Moreover, recent research indicates distillation remains more effective than RL for smaller models. In this work, we demonstrate that large-scale RL can significantly enhance the reasoning capabilities of strong, small- and mid-sized models, achieving results that surpass those of state-of-the-art distillation-based models. We systematically study the RL training process through extensive ablations and propose a simple yet effective approach: first training on math-only prompts, then on code-only prompts. Notably, we find that math-only RL not only significantly enhances the performance of strong distilled models on math benchmarks (e.g., +14.6% / +17.2% on AIME 2025 for the 7B / 14B models), but also code reasoning tasks (e.g., +6.8% / +5.8% on LiveCodeBench for the 7B / 14B models). In addition, extended code-only RL iterations further improve code benchmark performance with minimal or no degradation in math results. We develop a robust data curation pipeline to collect challenging prompts with high-quality, verifiable answers and test cases to enable verification-based RL across both domains. Finally, we identify key insights, including curriculum learning with progressively increasing response lengths and the stabilizing effect of on-policy parameter updates. We find that RL not only elicits the foundational reasoning capabilities acquired during pretraining and supervised fine-tuning (e.g., distillation), but also pushes the limits of the model's reasoning ability, enabling it to solve problems that were previously unsolvable.[2]

## 1 Introduction

Reasoning capabilities are a fundamental component of AI. Since the introduction of OpenAI o1 (OpenAI, 2024), building reasoning models using large-scale reinforcement learning (RL) has attracted significant attention. Remarkable progress has followed the open-sourcing of DeepSeek-R1 (Guo et al., 2025), empowering the open LLM and research communities to develop state-of-the-art reasoning models through RL or distillation. However, key technical details necessary for reproduction, such as data curation strategies and the specific RL training recipe, were omitted from the original DeepSeek-R1 report (Guo et al., 2025), leaving the community scrambling to replicate its success.

---

[*]Equal contribution. Correspondence to: `{yachen, zhuoliny, wping}@nvidia.com`

[2]We release the model, dataset, and a full technical report at: `https://huggingface.co/papers/2505.16400`.

39th Conference on Neural Information Processing Systems (NeurIPS 2025).

Subsequent efforts by different teams explored different model sizes (e.g., 1.5B (Luo et al., 2025b), 7B (Wen et al., 2025), 14B (Luo et al., 2025a), and 32B-only (Yu et al., 2025)), different initial checkpoints (e.g., base models (Yu et al., 2025) and distilled reasoning models (He et al., 2025)), and different target domains (e.g., math (Luo et al., 2025b), code (Luo et al., 2025a), and physical AI (Azzolini et al., 2025)). Each study shows a potential path to success in specific settings but lacks a conclusive or consistent training recipe. Moreover, both DeepSeek-R1 (Guo et al., 2025) and Llama-Nemotron (Bercovich et al., 2025) report that distillation outperforms RL for small and mid-sized models, recommending RL only for the largest models, such as the DeepSeek-V3-671B (Liu et al., 2024a) or Llama-Nemotron-Ultra-253B — a strategy also adopted by the Qwen3 (Qwen, 2025a).

In this work, we demonstrate that large-scale reinforcement learning (RL) can significantly enhance the reasoning capabilities of strong small- and mid-sized SFT models (DeepSeek-R1-Qwen-Distilled-7B/14B) - achieving performance competitive with state-of-the-art distillation-based results at 7B, and surpassing them at 14B (Moshkov et al., 2025; Ahmad et al., 2025).

Specifically, we make the following contributions:

1. We propose conducting math-only and code-only RL separately: the distilled SFT model is first trained on math-only prompts, followed by training on code-only prompts. This approach was initially motivated by training efficiency considerations, as the average verification time for code is significantly longer than that for math. Subsequently, we found two exciting observations: *i)* Math-only RL significantly boosts the performance of strong distilled models not only on math benchmarks (e.g., +14.6% / +17.2% on AIME 2025 for the 7B / 14B models), but also on code reasoning tasks (e.g., +6.8% / +5.8% on LiveCodeBench v5 for the 7B / 14B models); see Table 1 for details. *ii)* Extended iterations of code-only RL lead to minimal or no degradation on math reasoning tasks (e.g., +1.0% / -0.8% on AIME 2024 / 2025 for the 7B model); see Table 4 for details. These observations contrast with domain-specific supervised fine-tuning (SFT), which can lead to catastrophic forgetting and degraded performance on other domains.

2. We develop and share a systematic data curation recipe to collect high quality math problems with verifiable answers, as well as coding descriptions with test cases, ensuring that all data is reliable and testable. We open-source the dataset for the benefit of the community at https://huggingface.co/datasets/nvidia/AceReason-Math

3. To ensure consistent conclusions, we examine the RL training recipe through detailed ablation studies and analysis under state-of-the-art settings. Our findings include: *i)* Curriculum learning with a progressively increasing maximum response length improves both training efficiency and final accuracy on reasoning benchmarks. *ii)* On-policy parameter updates stabilize the RL process. *iii)* RL not only elicits the foundational reasoning capabilities acquired during pretraining and supervised fine-tuning (e.g., distillation), as evidenced by significant improvements in pass@1, but also expands the model's capabilities to solve previously unsolvable problems, as demonstrated by substantial and consistent gains from pass@64 to even pass@1024.

## 2 Related Work

Training LLMs to reason has been a long-standing research focus (Wei et al., 2022), especially in the domains of code (Chen et al., 2021) and math (Cobbe et al., 2021). In recent years, major development efforts have focused on acquiring reasoning capabilities by training on math and code data during both the pretraining and supervised fine-tuning (SFT) stages (Shao et al., 2024; Guo et al., 2024; Grattafiori et al., 2024; Yang et al., 2024b; Liu et al., 2024d). Reinforcement learning (RL) has previously been explored for math reasoning using reward models tailored to the math and code domains (Shao et al., 2024; Yang et al., 2024b). However, the gains have been limited, largely due to the inherent challenges of reward modeling in mathematical and coding domains (Lightman et al., 2023; Liu et al., 2024d,b).

The release of OpenAI o1 (OpenAI, 2024), and especially the open-sourcing of DeepSeek-R1 (Guo et al., 2025), highlights the effectiveness of large-scale RL through rule-based verification. In the case of math problems with deterministic answers, models are required to output the final result in a specific format (e.g., boxed), enabling accurate rule-based verification (e.g., Yang et al., 2024b; Liu

et al., 2024d). For code problems, feedback is provided through compilation and execution against predefined test cases (e.g., Zeng et al., 2025; Luo et al., 2025a).

Due to the absence of key implementation details in frontier models, such as RL training recipes and data curation strategies, subsequent works have explored and shared data curation methods (Luo et al., 2025b; He et al., 2025; Luo et al., 2025a), and introduced various techniques to improve and stabilize the widely adopted GRPO training (Shao et al., 2024). These include progressively increasing the maximum response length (Luo et al., 2025b,a; He et al., 2025), clip-higher to mitigate entropy collapse (Yu et al., 2025), and overlong filtering to avoid penalties from truncated generations within the maximum response length (Yu et al., 2025). Many of these efforts focus exclusively on either the math domain (Luo et al., 2025b; Yu et al., 2025; Chen et al., 2025; RL Lab, 2025) or the code domain (Luo et al., 2025a; Zeng et al., 2025), highlighting the difficulty of handling heterogeneous prompts and inherent complexity of RL training. Furthermore, the range of reported benchmarks remains limited, typically to AIME 2024 / 2025, and LiveCodeBench (Jain et al., 2024), compared to broader evaluations in frontier reasoning models (Guo et al., 2025; Qwen, 2025a).

Another line of follow-up work focuses on distilling existing frontier reasoning models, which are originally trained via RL (Guo et al., 2025; Qwen, 2025b), through strong-to-weak distillation using rejection sampling (Ahmad et al., 2025; Moshkov et al., 2025; Bercovich et al., 2025), as prior studies have found that RL yields suboptimal results for smaller models compared to distillation (Guo et al., 2025; Bercovich et al., 2025). In this work, we initiate RL from strong distilled models, and show that it can achieve results that are competitive with or surpass existing state-of-the-art distillation-based approaches on math (Moshkov et al., 2025) and code (Ahmad et al., 2025).

# 3 Method

## 3.1 Framework

We adopt the GRPO algorithm (Shao et al., 2024), as used in DeepSeek-R1, instead of PPO (Schulman et al., 2017), due to its simplicity and the advantage of not requiring a separate value function model. For each question-answer pair $(q, a)$, we sample from policy model $\pi_{\theta_{\text{old}}}$ to generate a group of $G$ individual rollouts $\{o_i\}_{i=1}^G$. We assign a reward score $S_i = S(o_i, a)$ to each response $o_i$, given the oracle answer $a$, using a rule-based reward function $S$. We employ the token-level policy gradient loss variant of GRPO, as introduced by Yu et al. (2025),

$$\mathcal{J}_{\text{GRPO}}(\theta) = \mathbb{E}_{(q,a)\sim\mathcal{D},\ \{o_i\}_{i=1}^G \sim \pi_{\theta_{\text{old}}}(\cdot|q)}$$
$$\left[ \frac{1}{\sum_{i=1}^G |o_i|} \sum_{i=1}^G \sum_{t=1}^{|o_i|} \left( \min\left( r_{i,t}(\theta)\hat{A}_{i,t},\ \text{clip}\left(r_{i,t}(\theta), 1-\varepsilon, 1+\varepsilon\right)\hat{A}_{i,t} \right) - \beta D_{\text{KL}}(\pi_\theta||\pi_{\text{ref}}) \right) \right], \tag{1}$$

therein,

$$r_{i,t}(\theta) = \frac{\pi_\theta(o_{i,t} \mid q, o_{i,<t})}{\pi_{\theta_{\text{old}}}(o_{i,t} \mid q, o_{i,<t})},\ \hat{A}_{i,t} = \frac{S_i - \text{mean}(\{S_i\}_{i=1}^G)}{\text{std}(\{S_i\}_{i=1}^G)} \text{ for } \forall t, \tag{2}$$

where $r_{i,t}(\theta)$ is the token-level importance weight, and the token-level advantage $\hat{A}_{i,t}$ within each response is uniformly assigned with the value of normalized reward score across the group $\{S_i\}_{i=1}^G$.

Our experiments strictly adhere to on-policy training by performing a single gradient update after generating a group of $G$ rollouts. This approach ensures stable RL training and helps prevent entropy collapse. Consequently, the policy used for data collection matches the current policy, i.e., $\pi_{\theta_{\text{old}}}(\cdot \mid q) = \pi_\theta(\cdot \mid q)$, and the importance weight $r_{i,t}(\theta) = 1$. Additionally, we eliminate the KL divergence term by setting $\beta = 0$, thus the GRPO objective became *REINFORCE* objective (?) with group-normalized rewards,

$$\mathcal{J}_{\text{GRPO}}(\theta) = \mathbb{E}_{(q,a)\sim\mathcal{D},\ \{o_i\}_{i=1}^G \sim \pi_\theta(\cdot|q)} \left[ \frac{1}{\sum_{i=1}^G |o_i|} \sum_{i=1}^G \sum_{t=1}^{|o_i|} \hat{A}_{i,t} \right]. \tag{3}$$

So, the update rule becomes,

$$\nabla_\theta \mathcal{J}_{\text{GRPO}}(\theta) = \mathbb{E}_{(q,a)\sim\mathcal{D}, \{o_i\}_{i=1}^G \sim \pi_\theta(\cdot|q)} \left[ \frac{1}{\sum_{i=1}^G |o_i|} \sum_{i=1}^G \sum_{t=1}^{|o_i|} \nabla_\theta \log \pi_\theta(o_{i,t} \mid q, o_{i,<t}) \cdot \hat{A}_{i,t} \right]. \tag{4}$$

We started RL experiments from distilled reasoning models DeepSeek-R1-Qwen-Distilled-7B/14B (Guo et al., 2025), ensuring that the experiments were conducted under controlled conditions without introducing variations in distillation data or fine-tuning compute. We use the veRL framework (Sheng et al., 2024), which implements token-level loss for GRPO, and employ the vLLM inference engine (v0.7.3) (Kwon et al., 2023) for sample generation. Our custom modification includes adding math and code reward functions (verifiers) to the implementation. All experiments are conducted using 128 NVIDIA H100 GPUs.

**Reward functions:**

- For verification of math problems, we employ a rule-based Python verification function built on top of `sympy`, following the approach of AceMath (Liu et al., 2024d). Specifically, it relies on antlr4-python3-runtime (v4.11.1) and sympy (v1.12). This configuration is crucial for ensuring accurate symbolic equivalence. We extract the answer from `\\boxed{}` appearing after the `<\think>` token and assign rewards strictly based on the correctness of this answer (1 for correct, 0 for incorrect), without applying any format-based rewards or length penalties. Using a process pool with 64 workers, the average verification time is approximately 3.9 seconds per 1,024 instances.

- For coding problem verification, we utilize a local sandbox verifier, following the code execution tools implemented in the LiveCodeBench repository (Jain et al., 2024). Given the model's output, we extract the code generated within ` ```python[code]``` ` code block that follows `<\think>` token. Binary rewards are then assigned based on code execution outcome on full set of test cases. A positive reward will be granted if and only if the extracted code successfully passes all test cases within the specific time limit. Using a process pool with 64 workers, the average verification time for code is approximately 552.4 seconds per 1,024 instances.

Given the significant difference in verification time between math and code, we propose conducting math-only and code-only RL separately.

### 3.2 Math-only RL

#### 3.2.1 Data Curation

We developed a data collection and verification pipeline to generate high-quality mathematical data for RL training. Our dataset combines DeepScaler (Luo et al., 2025b; Gao et al., 2024; Min et al., 2024) and NuminaMath (Li et al., 2024), covering algebra, combinatory, number theory, and geometry. We apply 9-gram filtering to avoid contamination with common math benchmarks and implement filtering rules to exclude unsuitable data, such as questions involving multiple sub-questions, multiple-choice or true/false questions, overly long or complex answers, proof-based questions, non-English content, references to figures, or excessively brief prompts.

Since NuminaMath data originates from online sources processed through OCR and parsing tools, it contains considerable noise due to incorrect questions or answers. To address this, we use the DeepSeek-R1 model with up to eight attempts per question, retaining only those that achieve correct majority-voted solutions via a rule-based verifier. Questions that are consistently unsolvable by DeepSeek-R1 often exhibit ambiguity or OCR-related errors upon human inspection and are therefore discarded. We further filter out questions requiring fewer than 2,000 R1 response tokens to answer, as we consider these questions to be solvable without extensive reasoning, and downsample problems with responses of 2,000–4,000 tokens to balance the dataset based on response length. Our final, rigorously verified dataset contains approximately 49,000 high-quality math problems for RL training.

#### 3.2.2 Training Process

RL training can be computationally intensive when involving long CoT reasoning, with around 80% training time spent on generating model outputs. To address this challenge, our RL pipeline focuses on enhancing reliability and efficiency through three primary strategies: 1) strict on-policy training to

maintain stable training and prevent entropy collapse, 2) stage-wise length extension from 8K to 32K tokens, and 3) curriculum training using increasingly difficult prompts at later stages.

- **On-policy training to stabilize entropy loss**. The entropy of the policy model serves as a key metric for assessing its ability to explore during RL training. In early experiments, we found applying multiple (2 or 4) gradient updates after model generation with a group of $G$ rollouts per prompt led to rapid entropy collapse around 100 steps (see Figure 2c). In contrast, using exactly one gradient update after model generation, as in original DeepSeek-Math's GRPO implementation (Shao et al., 2024), consistently prevented collapse. We therefore adopted this strict on-policy approach throughout RL training.

- **Length extension to accelerate training**. Length extension has been shown to be effective for smaller models (e.g., the 1.5B DeepScaler (Luo et al., 2025b)), but Wen et al. (2025) reported challenges in scaling to larger models, as training at an 8K response length initially led to degraded performance. In contrast, we were surprised to observe substantial performance improvements when extending training from 8K to 16K maximum response length. Based on this, we adopted a stage-wise length extension strategy ($8K \rightarrow 16K \rightarrow 24K \rightarrow 32K$) to enable more efficient training, as directly starting from 16K or 24K resulted in suboptimal results (see Figure 2b).

- **Harder problems to push the model**. We used curriculum learning by introducing more difficult prompts during the 24K and 32K stages. As the model mastered easier examples, their advantage reach 0 in the GRPO objective. We filtered prompts by model pass rate, filtering out those with pass rate $> 6/16$, which significantly improves model performance (Table 3).

**Math RL improves code reasoning**. In Table 1, we show performing math RL not only improves math reasoning on AIME24/25 but also boosts Live-CodeBench v5 score to 44.4% (6.8% ↑) for 7B and 58.9% (5.8% ↑) for 14B, which already outperforms the very recent code RL model DeepCoder-14B (57.9%) (Luo et al., 2025a). Furthermore, we show that

Table 1: Math-only RL improves code reasoning.

| Models | AIME24 avg@64 | AIME25 avg@64 | LCB v5 avg@8 |
|---|---|---|---|
| DeepSeek-R1-Distill-Qwen-7B | 55.5 | 39.0 | 37.6 |
| AceReason-7B (Math-only) | **69.0** | **53.6** | **44.4** (6.8↑) |
| DeepSeek-R1-Distill-Qwen-14B | 69.7 | 50.2 | 53.1 |
| AceReason-14B (Math-only) | **78.6** | **67.4** | **58.9** (5.8↑) |

math-only RL improves coding performance across all problem topics—not just math-related coding tasks (see Figure 9 in section A.8). This cross-domain generalization is a compelling advantage of reinforcement learning. In contrast, domain-specific supervised fine-tuning (SFT) often results in poor performance on other domains.

We used a batch size of 128, sampling $G = 8$ responses per prompt for 8K length training and 16 responses otherwise. We adopted a learning rate of $1 \times 10^{-6}$ with AdamW (Kingma, 2014), and set both the entropy loss coefficient and KL loss coefficient $\beta$ to 0.

## 3.3 Code-only RL

### 3.3.1 Data Curation

We curated our code-only RL training dataset from modern competitive programming platforms using strict selection criteria to ensure high-quality coding problems. The dataset includes both function-calling and standard input/output (stdin/stdout) formats and covers a wide range of algorithmic topics, including graph theory, data structures, number theory, greedy algorithms, and more.

To ensure stability for RL training, we filtered out problems incompatible with standard output comparison (e.g., multi-solution or interactive problems requiring special judges) or those needing platform-specific templates, thereby minimizing potential *false negative* reward. Furthermore, we curated strong testcases covering tricky edge cases or extreme cases under input limitations, ensuring that incorrect solutions would fail and thereby eliminating potential *false positive* reward. As discussed in Appendix A.9, both false positive reward and false negative reward can obfuscate RL training by introducing noisy reward signals. To gauge difficulty, we evaluated each problem using DeepSeek-R1-671B with 8 rollouts, assigning a difficulty score from 0 to 8. Problems where the model failed all 8 attempts (level 8) were excluded. Finally, we performed careful benchmark decontamination and problem deduplication across platforms using n-gram context analysis and

original URL matching (see Appendix A.7 for details). After such aggressive filtering process, $8,520$ problems remained, forming our final training set.

### 3.3.2 Training Process

We apply the two-stage code-only RL pipeline designed to accommodate models of varying scales. The pipeline leverages training sets composed of coding problems within specific difficulty ranges, along with customized settings for maximum response length and sampling temperature.

- **Stage 1** initiates the code RL process, launching after prior math-only RL to ensure training stability. In Stage 1, training data is constructed by difficulty: problems with difficulty up to level 5 are used for 7B model, while problems up to level 7 are used for 14B model. We set maximum response length as 24,000, temperature as 0.6 and number of rollouts as 8 for Stage 1 training.

- **Stage 2** employs the full set of training problems with 32,768 maximum response length. In this stage, we implement an epoch-wise filtering strategy by filtering out relatively easy problems w.r.t. prior epoch checkpoints and gradually increasing the sampling temperature from 0.6 to 1.0, number of rollouts from 8 to 16 across epochs. This aims to encourage policy convergence while encouraging exploration.

We set batch size to 128 and learning rate to $5 \times 10^{-6}$ with AdamW, continuing training in both stages until policy converges. For reward function, we adopt the strict rule-based reward: positive reward 1 is granted if and only if the generated code successfully passes all test cases for the given problem. As for efficient evaluation, we deploy a parallelized local verifier to check testcase correctness.

## 3.4 Summary of Training Curriculum

We use DeepSeek-R1-Distill-Qwen2.5-7B and 14B as our initial SFT models. To integrate math-only and code-only RL, we first perform math-only RL with stage-wise length extension from 8K to 24K. Next, we apply code-only RL, extending the length from 24K to 32K. Finally, we conduct math-only RL at 32K. We find that this training curriculum is slightly more effective and efficient in practice than first performing math-only RL from 8K to 32K, followed by code-only RL from 24K to 32K.

# 4 Evaluation

## 4.1 Experimental Setup

Our experiments start from strong SFT models, DeepSeek-R1-Distill-Qwen-7B and 14B, which are based on the Qwen2.5 model family (Yang et al., 2024a) and distilled from DeepSeek-R1 (Guo et al., 2025). To ensure consistency and reproducibility, we follow the DeepSeek-R1 evaluation protocol, using a temperature of $0.6$, `top-p` of $0.95$, and a maximum output length of 32,768 tokens.

### 4.1.1 Math Evaluation

We use a diverse math competition benchmarks, including AIME2024, AIME2025, MATH500 (Hendrycks et al., 2021), in addition with HMMT2025 Feb and BRUMO2025 from MathArena (Balunović et al., 2025). Due to the high variance in outputs from reasoning models when using sampling, we report pass@1 performance averaged over $k$ generations (avg@$k$). For small-scale benchmarks such as AIME, we use $k = 64$, following DeepSeek-R1. This choice of $k$ is critical for obtaining a reliable evaluation, as lower values of $k$ lead to a significantly higher standard error of the mean (e.g., on AIME2024 @16/32/64 : 1.8/1.2/0.7).

To isolate the effects of pre-training, we primarily compare with reasoning models based on either Qwen2.5 or Llama-3.1 at similar parameter scales. These include SFT models (distilled from much larger frontier models) such as Light-R1-7B (Wen et al., 2025), OpenMathReasoning-7/14/32B (Moshkov et al., 2025), and LLaMA-Nemotron-Nano/Super-8/49B (Bercovich et al., 2025), as well as RL models like AReal-boba-RL-7B (RL Lab, 2025), Skywork-OR1-Math-7B (He et al., 2025), and Light-R1-14B (Wen et al., 2025). For context, we also include frontier reasoning models such as DeepSeek-R1 (Guo et al., 2025), QwQ-32B (Qwen, 2025b), LLaMA-Nemotron-Ultra-253B (Bercovich et al., 2025), and o3-mini (OpenAI, 2024).

Table 2: **Math and Code reasoning evaluation**. We report pass@1 averaged over $k$ generations (avg@$k$) following the DeepSeek-R1 evaluation framework (template, temperature=0.6, `top_p`=0.95, max response length=32,768). By default, we report self-reported numbers from model developers if they are available. Otherwise, [†]we evaluate the model using the same evaluation setting, or [‡]we collected from MathArena or LiveCodeBench leaderboard.

| Models | AIME 2024 avg@64 | AIME 2025 avg@64 | MATH 500 avg@4 | HMMT 2025 avg@64 | BRUMO 2025 avg@64 | LiveCodeBench v5 avg@8 | LiveCodeBench v6 avg@8 | Codeforces ELO pass@1 | Codeforces Percentile pass@1 | EvalPlus avg@4 |
|---|---|---|---|---|---|---|---|---|---|---|
| QwQ-32B | 79.5 | 65.8[‡] | 96.0 | 47.5[‡] | – | 63.4 | – | 1982 | 97.7 | – |
| DeepSeek-R1-671B | 79.8 | 70.0[‡] | 97.3 | 41.7[‡] | 80.8[‡] | 65.9 | – | 2029 | 98.1 | – |
| Llama-Nemotron-Ultra-253B | 80.8 | 72.5 | 97.0 | – | – | 66.3 | – | – | – | – |
| o3-mini (low) | 60.0 | 48.3[‡] | 95.8 | 28.3[‡] | 66.7[†] | 60.9[‡] | – | 1918 | 97.1 | – |
| o3-mini (medium) | 79.6 | 76.7[‡] | 97.3 | 53.3[‡] | 80.0[†] | 67.4[‡] | – | 2036 | 98.1 | – |
| AReal-boba-RL-7B | 61.9 | 48.3 | 93.8[†] | 29.4[†] | 58.9[†] | 34.3[†] | – | – | – | – |
| Skywork-OR1-Math-7B | 69.8 | 52.3 | 94.4[†] | 31.4[†] | 60.6[†] | 43.6 | – | – | – | – |
| OlympicCoder-7B | – | – | – | – | – | 40.7 | 37.1[†] | – | – | 79.8[†] |
| Light-R1-7B | 59.1 | 44.3 | 92.4[†] | 27.6[†] | 52.8[†] | 40.6[†] | 36.4[†] | – | – | – |
| Light-R1-14B | 74.0 | 60.2 | 94.6[†] | 37.8[†] | 67.1[†] | 57.9[†] | 51.5[†] | – | – | – |
| DeepCoder-14B (32K-Inference) | 71.0[†] | 56.1[†] | – | – | – | 57.9 | 50.4[†] | 1922 | 97.2 | 85.3[†] |
| OpenMath-Nemotron-7B | 74.8 | 61.2 | – | – | – | – | – | – | – | – |
| OpenMath-Nemotron-14B | 76.3 | 63.0 | – | – | – | – | – | – | – | – |
| OpenMath-Nemotron-32B | 76.5 | 62.5 | – | – | – | – | – | – | – | – |
| OpenCodeReasoning-Nemotron-7B | – | – | – | – | – | 51.3 | 46.1[†] | – | – | 83.4[†] |
| OpenCodeReasoning-Nemotron-14B | – | – | – | – | – | 59.4 | 54.1[†] | – | – | 84.1[†] |
| Llama-Nemotron-Nano-8B-v1 | 61.3 | 47.1 | 95.4 | – | – | 46.6 | 46.2[†] | – | – | 81.2[†] |
| Llama-Nemotron-Super-49B-v1 | 67.5 | 60.0 | **96.6** | – | – | 45.5 | – | – | – | – |
| DeepSeek-R1-Distill-Qwen-7B | 55.5 | 39.0[†] | 92.8 | 26.3[†] | 51.2[†] | 37.6 | 34.1[†] | 1189 | 57.4 | 80.4[†] |
| DeepSeek-R1-Distill-Qwen-14B | 69.7 | 50.2[†] | 93.9 | 31.7[‡] | 61.1[†] | 53.1 | 47.9[†] | 1481 | 85.6 | 83.9[†] |
| DeepSeek-R1-Distill-Qwen-32B | 72.6 | 54.9[†] | 94.3 | 33.3[‡] | 68.3[‡] | 57.2 | – | 1691 | 93.2 | – |
| DeepSeek-R1-Distill-Llama-70B | 70.0 | 55.0[‡] | 94.5 | 33.3[‡] | 66.7[‡] | 57.5 | – | 1633 | 91.4 | – |
| AceReason-Nemotron-7B | 69.0 | 53.6 | 94.1 | 33.9 | 62.2 | 51.8 | 44.1 | 1475 | 84.8 | 84.6 |
| AceReason-Nemotron-14B | **78.6** | **67.4** | 95.0 | **46.4** | **72.3** | **61.1** | **54.9** | **2024** | **98.1** | **85.7** |

## 4.1.2 Code Evaluation

For coding tasks, we evaluate our AceReason-Nemotron models on LiveCodeBench (LCB) (Jain et al., 2024) v5 ($20240801 − 20250201$) and v6 ($20250201 − 20250501$) subsets, containing recently released AtCoder, LeetCode problems. We also report Codeforces ELO and percentile number of our models based on LiveCodeBench Pro dataset (Zheng et al., 2025), which contains Codeforces problems from 202407 to 202412. We also include evaluations on EvalPlus (Liu et al., 2024c, 2023).

We compare our model with state-of-the-art open-sourced code-gen LLMs of similar parameter scales, including OlympicCoder-7B (Face, 2025), Llama-3.1-Nemotron-Nano-8B-v1 (Bercovich et al., 2025), OpenCodeReasoning-7B/14B (Ahmad et al., 2025), DeepCoder-14B (Luo et al., 2025a). For further context, we also include strong frontier reasoning models as titled above.

## 4.2 Main Results

From the evaluation results in Table 2, we summarize the key conclusions as follows:

- **RL significantly improves reasoning capabilities.** Our AceReason-Nemotron-7B/14B models show that using RL significantly improves over the initial SFT models (DeepSeek-R1-Distill-Qwen-7B/14B) on both math and coding tasks. Specifically, for math tasks, our AceReason-Nemotron-7B achieves remarkable improvements over SFT model by increasing $14.5\%$ accuracy on AIME 2024, and $14.6\%$ accuracy on AIME 2025. For coding tasks, it achieves $14.2\%$ and $8\%$ accuracy improvements over DeepSeek-R1-Distill-Qwen-7B on LiveCodeBench v5 and v6, respectively. Meanwhile, AceReason-Nemotron-14B improves from $69.7\%/50.2\%$ to $78.6\%/67.4\%$ on AIME24/25, and $53.1\%/47.9\%$ to $61.1\%/54.9\%$ on LiveCodeBench v5/v6 from initial SFT model DeepSeek-R1-Distill-Qwen-14B, even surpassing significantly larger SFT models such as DeepSeek-R1-Distill-Qwen-32B and DeepSeek-R1-Distill-Llama-70B. Additional results on 1.5B and 32B further validate the same improvements as shown in Appendix Table **??**.

- **AceReason-Nemotron vs. SOTA RL-based reasoning models.** While comparing with state-of-the-art open RL-based reasoning models under the same parameter scale, AceReason-Nemotron model still remains its superiority. In math reasoning domain, AceReason-Nemotron-7B model provides competitive results while comparing with strong RL-based models (Skywork-OR1-Math, Light-R1, etc.), while AceReason-Nemotron-14B provides the best-in-class results. In code

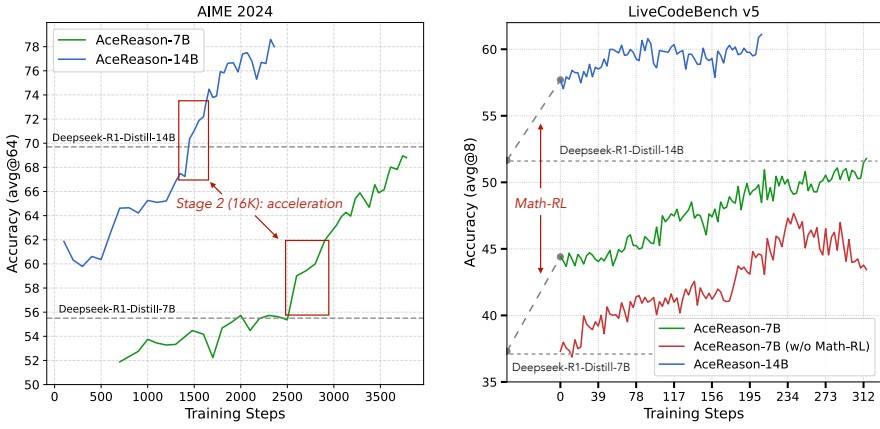

Figure 1: Model accuracy on AIME2024 and LiveCodeBench v5 during RL.

generation domain, AceReason-Nemotron models outperform all open-sourced reasoning models with similar parameter scale. Its Math-Code ability remains competitive even comparing with frontier reasoning models, such as QWQ-32B, o3-mini, etc.

- **AceReason-Nemotron vs. SOTA models through distillation.** AceReason-Nemotron-14B shows better math and code performance than the latest SOTA specialized distilled model such as OpenMath-14B/32B by +2.1%/+4.4% on AIME24/25, OpenCodeReasoning-14B by +1.7%/+0.8% on LiveCodeBench v5/v6, demonstrating RL leads to higher upper-bound of model performance than distillation. In contrast, at 7B model scale, AceReason-Nemotron-7B performs competitively with OpenCodeReasoning-7B on coding tasks, while OpenMath-7B appears to have bigger advantage than RL-trained AceReason-Nemotron-7B on math reasoning. In all cases, AceReason-Nemotron models significantly outperform the general-purpose reasoning model Llama-Nemotron-8B/49B, which is trained via distillation. This indicates that the effectiveness of distillation versus RL still depends on model size and task domain, though RL offers the potential for significantly higher accuracy at the 14B scale and beyond.

## 4.3 Analysis

**Response length grows alongside performance.** Figure 1 (left subfigure) and Figure 2a show the evolution of response length on the training set and corresponding AIME24 accuracy (AIME25 in appendix Figure 6) throughout the RL training of AceReason-Nemotron-7B model. We analyze two distinct stages in the 8K → 16K length-extension training strategy: **1) Stage 1 (8K) – Transition from imitation to RL:** During this stage, the model learns to compress its reasoning process to adapt to an 8K token limit, causing an initial drop in problem-solving capabilities compared to baseline. However, after approximately 1K–2K RL steps, the model gradually recovers its performance; **2) Stage 2 (16K) – Accelerated learning:** Upon extending the token limit to 16K, the model immediately exploits the extra reasoning tokens capacity: within 500 RL steps, its average response length increases from 4K to around 6.5K tokens. At the same time, AIME24 accuracy improves sharply from $55\%$ to $62\%$, after which both response length and accuracy plateau. Extending the maximum response length further to 24K and 32K demonstrates a similar trend.

**Hard prompts drive the largest gains.** At the 24K response length RL stage, we build {Easy, Medium, Hard}-prompt sets based on difficulty estimated by 7B model's performance over 16 attempts. Easy prompt set includes prompts except those solved more than 14 times, Medium prompt set excludes those solved more than 10, and Hard excludes those solved more than 6. As shown in Table 3, our ablation confirms that training with Hard prompts yields a 2.6% improvement on the AIME24 benchmark com-

Table 3: Prompt difficulty and its impact on Stage 3 (24K) training.

| Data | AIME24 avg@64 | AIME25 avg@64 |
|---|---|---|
| Starting Checkpoint - 7B | 62.2 | 50.2 |
| **Full** (no filtering, #49K) | 63.3 | 51.1 |
| **Easy** (#10K) | 64.4 | 50.8 |
| **Medium** (#4.6K) | 65.3 | 51.9 |
| **Hard** (#2.2K) | **65.9** | **52.5** |

pared to fullset data and outperforms using Easy/Medium prompts, although it only has 2.2K prompts.

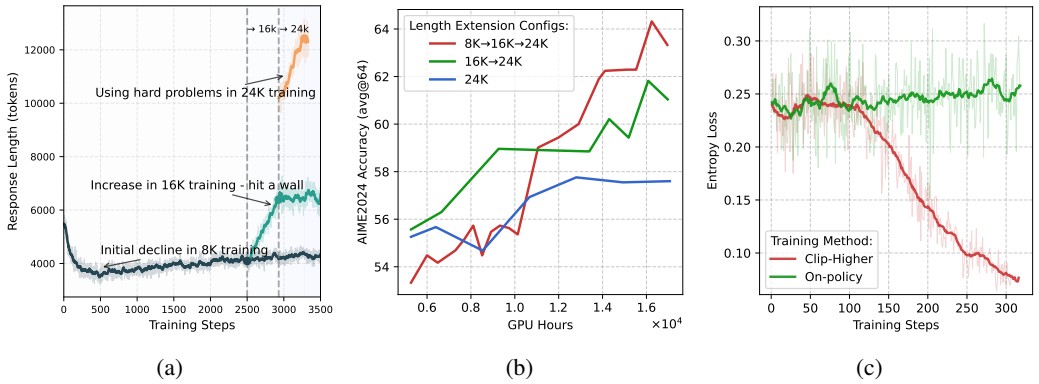

Figure 2: Analysis of RL training: (a) response length during math-RL training, (b) GPU hours for different length extension configurations, and (c) entropy of output logits using on-policy RL training versus off-policy training with Clip-Higher trick.

**Starting from 8K improves later performance.** Although training initially at 8K max response length offers faster training per step than starting at 16K or 24K, the early decline in AIME24 accuracy raises questions about its necessity for 7B-sized models. Figure 2b presents an ablation study comparing three strategies: 8K→16K→24K, 16K→24K, and directly at 24K. Each strategy utilizes identical full data, with accuracy measured on AIME24 alongside GPU hours consumed. The results indicate that while the 16K→24K strategy initially outperforms the baseline, the 8K→16K strategy quickly catches up during the 16K stage and consistently outperforms other strategies thereafter. Conversely, starting directly at 24K results in minimal performance gains and significantly longer training times, suggesting limited effectiveness during RL training for 7B model.

**Interplay of math-only RL and code-only RL.** In Figure 1 (right subfigure), we observe that initializing code RL training from math-RL checkpoints offers a substantially better starting point and results in significantly higher final accuracy on LCB. We investigate the effects of sequential Math RL and Code RL training stages on developing strong reasoning model in both math and code reasoning domains. Starting from DeepSeek-R1-Distill-Qwen-7B/14B as our initial models, we first apply Math-RL and then Code-RL, evaluating performance on standard Math (AIME 24/25) and Code (LCB v5/v6) benchmarks. Surprisingly, as shown in Table 4, initial Math-RL training not only significantly improves accuracy on math benchmarks, but also improves model performance on coding. Subsequent Code-RL training further boosts coding scores, with negligible degradation on math performance. This shows that the proposed sequential training strategy is effective in developing models with strong reasoning abilities across math and coding domains.

**Does RL improve pass@k or pass@1 over the distilled model?** Recent studies (Shao et al., 2024; Yue et al., 2025) suggest that RL primarily improves pass@1 accuray over SFT model (e.g., DeepSeek-R1-Distilled) without significantly impacting pass@$k$. However, Figure 3 demonstrates that RL consistently improves pass@$k$ score (from $k = 8$ to $k = 64$) for both

Table 4: Interplay of math-only RL and code-only RL.

| Models | AIME 2024 avg@64 | AIME 2025 avg@64 | LiveCodeBench v5 avg@8 | LiveCodeBench v6 avg@8 |
|---|---|---|---|---|
| DeepSeek-R1-Distill-Qwen-7B | 55.5 | 39.0 | 37.6 | 34.1 |
| + Math-RL (8K → 24K) | 65.9 (10.4↑) | 52.5 (13.5↑) | 44.4 (6.8↑) | 37.6 (3.5↑) |
| + + Code-RL (24K → 32K) | 66.9 (1.0↑) | 51.7 (0.8↓) | 51.8 (7.4↑) | 44.1 (6.5↑) |
| DeepSeek-R1-Distill-Qwen-14B | 69.7 | 50.2 | 53.1 | 47.9 |
| + Math-RL (8K → 24K) | 76.6 (6.9↑) | 63.4 (13.2↑) | 58.6 (5.5↑) | 50.9 (3.0↑) |
| + + Code-RL (24K → 32K) | 75.7 (0.9↓) | 63.9 (0.5↑) | 61.1 (2.5↑) | 54.9 (4.0↑) |

7B and 14B models on LiveCodeBench v5 and v6, with an improved pass@k scores maintaining a 10% margin from pass@8 to pass@64. On AIME24/25, we found the 7B and 14B model also show better pass@k compared to the SFT model across all $k$. For both AIME and LiveCodeBench, we generated 128 responses for each question and randomly sampled $k$ to calculate the pass@$k$ result with an average of 100 runs to reduce variance.

To further validate our conclusion, we extend pass@k evaluation from k= 64 to 1024 on Live-CodeBench v6, where correct answers are difficult to "guess" through limited sampling. For each question, we generated 1,024 responses to compute pass@1024. For smaller values of k (k < 1024), we randomly sampled k responses from the 1,024 and calculated pass@k by averaging over 100 runs. In Figure 5, we observe that our AceReason-Nemotron-7B consistently outperforms

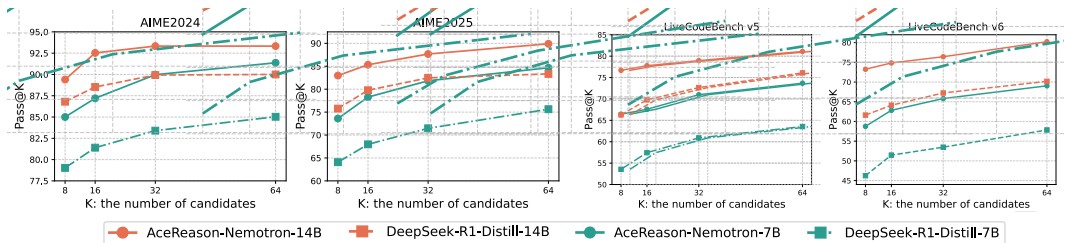

Figure 3: The Pass@K of RL (AceReason) and SFT (DeepSeek-R1-Distilled) models on AIME 2024/2025 and LiveCodeBench v5/v6.

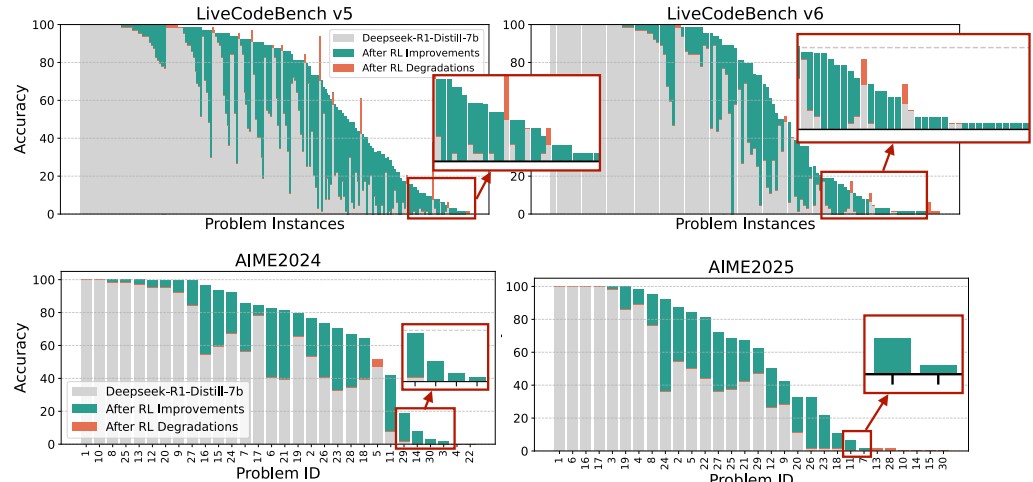

Figure 4: Problem-level solving rates comparison between distilled model and after RL training. Accuracy for each problem is calculated on average of 64 attempts.

the SFT model (DeepSeek-R1-Distill-Qwen-7B) by approximately 10% across all pass@$k$ values. Pass@K=1024 on AIEM25 shows a similar gap in Table 8.

**Where does RL improve over the distilled model?** Figure 4 compares the problem-level accuracies of the initial 7B SFT model with AceReason-Nemotron-7B after RL on LCB v5/v6 and AIME 2024/2025. Results for the 14B model are shown in Appendix Figure 8. On LCB, we observe RL unlocks a long tail of hard coding problems that the distilled model fails to solve in 64 attempts, adding 30 and 23 additional solvable problems to LCB v5 and v6. It also significantly improves on challenging problems where the SFT model has lower than 20% accuracy. On AIME, for the most challenging problems with zero solve rate, RL enables the model to solve 3 more problems on AIME24. Hence, we find RL not only improves the accuracy on problems with high solve-rate but also extends the boundary to solve hard problems that the SFT model was initially unable to solve.

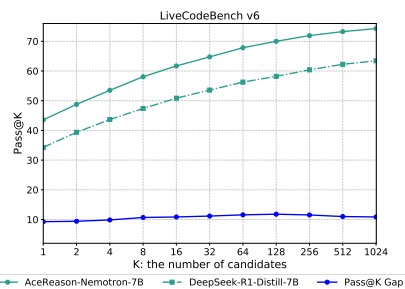

Figure 5: The Pass@K of RL (AceReason) and SFT model (DeepSeek-R1-Distilled) on LCB v6.

## 5 Conclusion

We demonstrate that large-scale RL can substantially enhance the reasoning capabilities of strong, small- and mid-sized distilled models. We propose performing RL on math-only prompts first, followed by code-only prompts. Notably, math-only RL significantly boosts performance not only on math benchmarks but also on code reasoning tasks. Crucially, subsequent code-only RL further improves code benchmark performance with minimal to no degradation in math results. To support this process, we develop a robust data curation pipeline that collects challenging prompts with high-quality, verifiable answers and test cases, enabling verification-based RL across both domains.

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

# A Technical Appendices and Supplementary Material

## A.1 Broader Impact and Limitations

This work primarily focuses on enhancing the reasoning capabilities of language models in mathematics and coding tasks. As such, it poses minimal negative societal impact in terms of privacy risks or bias concerns. Our study targets medium-scale models (7B/14B) due to the cost of RL training and extensive ablations; we additionally report 1.5B and 32B results in the appendix (Table 9), but do not evaluate models below 1B or above 100B parameters. We intentionally initialize from DeepSeek-R1–distilled checkpoints to isolate the effect of the SFT→RL stage; applying RL directly to base pre-trained models (e.g., R1-Zero–style) is a distinct setting and out of scope in this study. We focus on verifiable domains (math and coding) to enable rigorous evaluation; assessing transfer to open-ended tasks is left for future work. Finally, our conclusions are primarily empirical rather than theoretical.

## A.2 Funding Transparency Statement

1. Funding (financial activities supporting the submitted work): Funding in direct support of this work is from company NVIDIA;
2. Competing Interests (financial activities outside the submitted work): No additional revenues related to this work.

## A.3 Dataset

Table 5: Evaluation Dataset

| Dataset | # | License | Link |
|---|---|---|---|
| MATH500 (Lightman et al., 2023) | 500 | MIT | HuggingFaceH4/MATH-500 |
| AIME2024 | 30 | cc-by-nc-sa-4.0 | github.com/QwenLM/Qwen2.5-Math |
| AIME2025 | 30 | cc-by-nc-sa-4.0 | MathArena/aime_2025 |
| HMMT2025 Feb | 30 | cc-by-nc-sa-4.0 | MathArena/hmmt_feb_2025 |
| BRUMO2025 | 30 | cc-by-nc-sa-4.0 | MathArena/brumo_2025 |
| LiveCodeBench v5 (Jain et al., 2024) | 279 | MIT | livecodebench/code_generation_lite |
| LiveCodeBench v6 (Jain et al., 2024) | 175 | MIT | livecodebench/code_generation_lite |

Table 6: Training Dataset

| Dataset | License | Link |
|---|---|---|
| NuminaMath (Li et al., 2024) | Apache-2.0 | AI-MO/NuminaMath-1.5 |
| DeepScaler (Luo et al., 2025b) | MIT | agentica-org/DeepScaleR-Preview-Dataset |
| AtCoder Problems (before 20240731) | cc-by-4.0 | https://atcoder.jp/ |
| Aizu Problems | cc-by-4.0 | https://onlinejudge.u-aizu.ac.jp/ |
| LeetCode Problems | cc-by-4.0 | https://leetcode.com/ |

## A.4 Additional Experiments on General Reasoning

In addition to the math and code evaluation, we evaluate the models on STEM and logical reasoning benchmarks such as GPQA (Rein et al., 2024), Knight&Knave (Xie et al., 2024), and Zebra-Logic (Lin et al., 2025). In Table 7, we show that after RL, AceReason-Nemotron model improves on all datasets, especially on Knight&Knave with a large margin, achieving 96.1% with the 14B model. This shows the RL reasoning capability learned during math and code training generalizes to other domains.

## A.5 Additional Experiments on 1.5B and 32B models

In Table 9, we run the same AceReason-Nemtron RL recipe on a smaller and larger distilled checkpoint (DeepSeek-R1-Distilled-Qwen1.5/32B). We show that our training recipe generalizes to 1.5B model and 32B model, achieving significant improvements on both math and code benchmarks. Due to computational constraints, we only conduct Math-only RL for the 32B model. However, we found the coding accuracy on LCB v5 significantly improves to 63.4 without any code RL training, even surpassing Skywork-OR1-32B (63), which has done code RL.

Table 7: Benchmark accuracy (%) on GPQA, Knight & Knave, and Zebra-Logic.

| Model | GPQA | Knight & Knave | Zebra-Logic |
|---|---|---|---|
| DeepSeek-R1-Distill-7B | 49.1 | 35.2 | 24.5 |
| AceReason-Nemotron-7B | 51.1 | 56.4 | 39.6 |
| DeepSeek-R1-Distill-14B | 59.1 | 83.8 | 60.1 |
| AceReason-Nemotron-14B | 61.2 | 96.1 | 72.7 |

Table 8: AIME 2025 pass@k (%).

| Model | 8 | 16 | 32 | 64 | 128 | 256 | 512 | 1024 |
|---|---|---|---|---|---|---|---|---|
| DeepSeek-R1-Distilled-Qwen-7B | 62.2 | 66.8 | 71.3 | 76.5 | 81.2 | 86.2 | 90.3 | 92.6 |
| AceReason-Nemotron-7B | **73.4** | **76.0** | **78.3** | **85.4** | **89.0** | **93.0** | **94.8** | **96.1** |

## A.6 Additional Math-RL Analysis

In Figure 6, we show the accuracy on AIME2025 during math RL training. We found the stage 2 (16K) training leads to a large performance improvement within 300 steps. In Figure 7, we plot boxplots of AIME2024 and AIME2025 accuracy for the 7B and 14B AceReason model comparing to DeepSeek-R1-Distill-Qwen. We can observe the accuracy of AceReason over 64 generations at 25% and 75% percentile and median, significantly improve over the distilled model. Finally, in Figure 8, we inspect the problem-level solve rate for AceReason-14B on AIME2024 and AIME2025. On AIME24 and AIME25, the AceReason model solves one additional problem. We also observe large gains on problems with higher initial accuracy, showing the benefits of RL training.

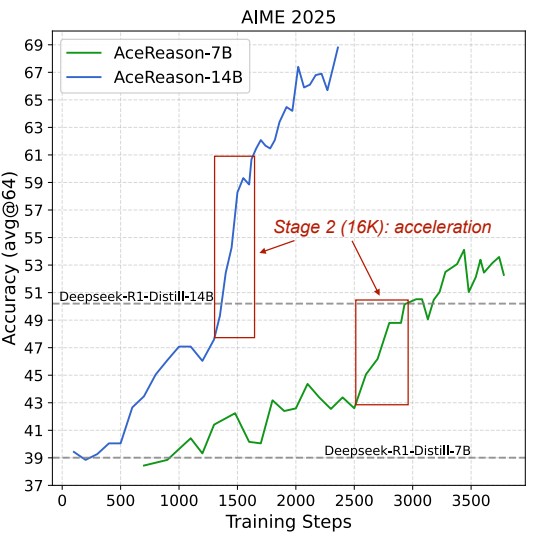

Figure 6: Model accuracy on AIME2025.

## A.7 Code-RL Dataset Curation Details

We collect our Code-RL training data from various modern competitive programming platforms, such as AtCoder, LeetCode, Aizu, etc., with public strong test cases, while most open-sourced coding datasets (e.g., TACO, APPs) suffer from noisy problem statements, self-contamination, and weak LLM synthetic test cases that are unreliable for RL training. To ensure data quality, we performed very strict filtering rules, by filtering out 1) Multi-solution or interactive problems that requires special judge or other external tools; 2) Problems where images within the statement obscure a clear understanding; 3) Problems containing incorrect test cases or those lacking golden solutions; 4) Problems with weak test cases that allow incorrect solutions to pass all tests. Furthermore, to prevent self-contamination within our collected problem set, we conduct strict problem statement and source

Table 9: Additional results of 1.5B and 32B model on AIME24, AIME25, and LiveCodeBench v5.

| Model | AIME24 | AIME25 | LiveCodeBench v5 |
|---|---|---|---|
| DeepSeek-R1-Distilled-Qwen-1.5B | 28.9 | 22.7 | 16.9 |
| AceReason-Nemotron-1.5B | 47.7 (18.8↑) | 35.5 (12.8↑) | 27.2 (10.3↑) |
| DeepSeek-R1-Distilled-Qwen-32B | 72.6 | 54.9 | 57.2 |
| AceReason-Nemotron-32B (math-only RL) | 79.0 (6.4↑) | 70.6 (15.7↑) | 63.4 (6.2↑) |

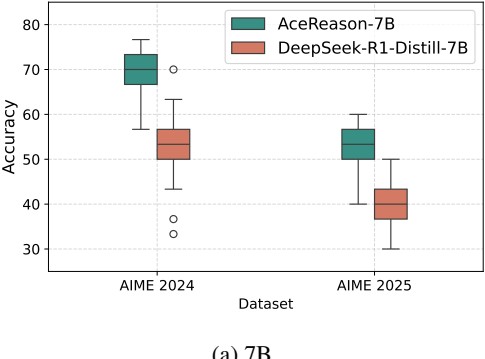

(a) 7B

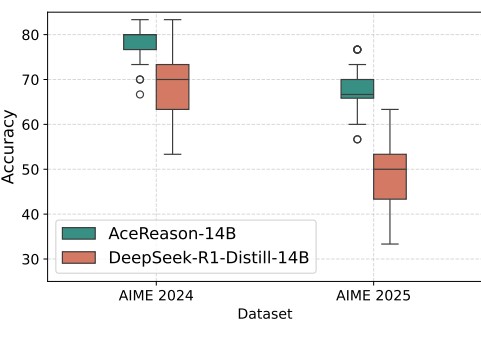

(b) 14B

Figure 7: Boxplot of AceReason vs Deepseek-R1-Distill on AIME24/25 over 64 generations.

URL matching. To avoid any potential contamination of our test set, we exclude all problems released after $20240801$, and apply $n$-gram matching (with $n = 14$) to our testing set problem statement.

To prepare for subsequent curriculum training for Code-RL, we estimate the relative difficulty of the collected problems. We deploy the local DeepSeek-R1-671B model, allow it to generate $8$ attempts on each problem, and assign a corresponding difficulty score ranging from $0$ to $8$. Problems that the DeepSeek-R1-671B model failed to solve in all $8$ attempts are excluded from the training set. Following this aggressive filtering process, $8,520$ problems remain, forming our final Code-RL training set.

### A.8 Topic-wise Accuracy Analysis on LiveCodeBench v5

Given the observation that both Math-RL and Code-RL enhance code generation ability on coding benchmarks, we are interested to see in detail how these two RL stages improve accuracy on topic-specific coding problems. Specifically, this ablation aims to identify which problem topics benefit the most from Math-RL and the subsequent Code-RL. Motivated by this, we conducted ablation studies on LiveCodeBench v5 dataset, which consists of coding problems from AtCoder and LeetCode platforms. While LeetCode problems come with human-annotated topic tags (e.g., Greedy, Math, DFS), there is no tag on Atcoder problems. To address this, we first extract a set of problem tags from LeetCode. Then, for each AtCoder problem, we query the o4-mini-high model to infer candidate tags given the problem statement and the set of topics. Furthermore, we group all LiveCodeBench v5 problems by their assigned topics and evaluate model performance (avg@8 accuracy) for each topic group.

We compare the performance of our initial SFT models, DeepSeek-R1-Distill-Qwen-7B/14B, against corresponding AceReason-Nemotron-7B/14B after applying Math-RL stage only and final models that incorporate both Math-RL and Code-RL. As shown in Figure 9, we plot the accuracy for each topic before and after Math-RL and Code-RL. The figure shows that applying math-only RL enhances model performance across all coding problem topics, with especially strong gains in algorithmic and math-related areas such as Math, Counting, and Combinatorics—domains that rely heavily on mathematical concepts and general reasoning abilities. Furthermore, for topics like Simulation, String, and Graph, which rely more heavily on coding implementation and data structure skills, Code-RL leads to significant further improvement.

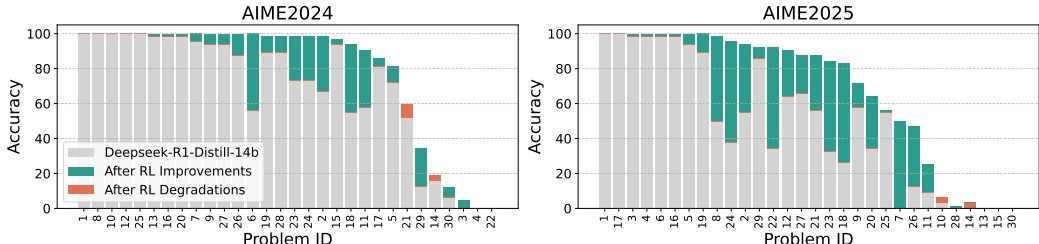

Figure 8: Comparison of problem-solving rates after RL training.

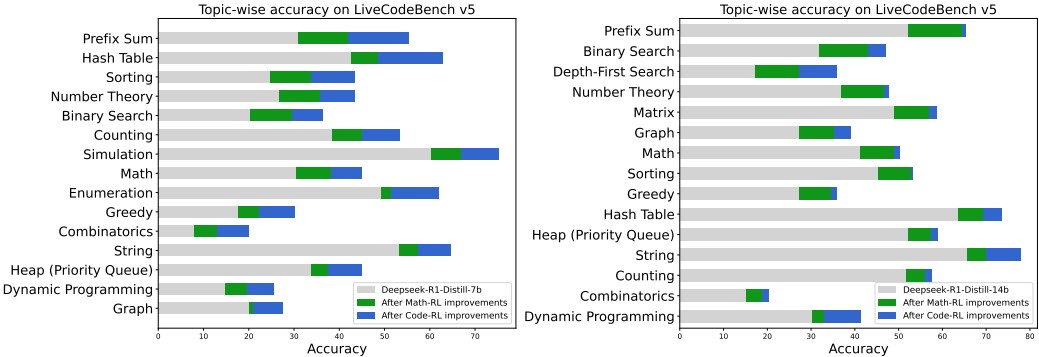

Figure 9: RL improvement on topic-level for coding problems.

## A.9 False Positives and Negatives in Code-RL Training

To highlight the importance of eliminating *false positive* reward (incorrect code passing all tests within time constraints) and *false negative* reward (incorrect test cases that fail correct code) in RL Training, we conduct two ablation experiments, showing that both types of errors can be harmful to RL training, resulting in early convergence on sub-optimal policies, or even complete training collapse.

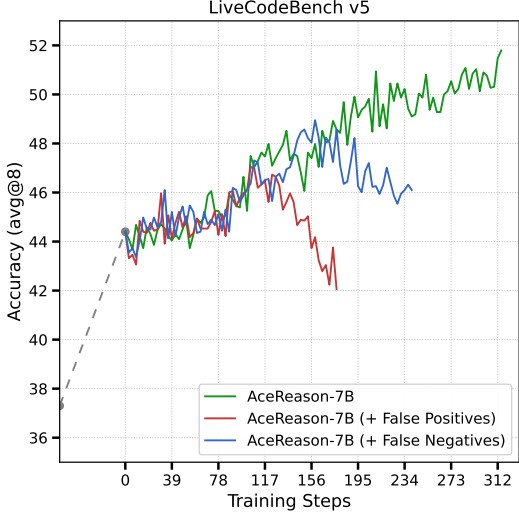

Figure 10: The impact of false positive and false negative rewards in Code RL Training

To simulate the impact of false negative rewards, we introduce a subset of problems into the training data where either the correctness of test cases could not be verified, or the official "golden" solution failed to pass all provided tests. From Figure 10 we can see that, the RL model tends to converge to a sub-optimal point, with no further improvement on the evaluation set. We hypothesize that such

false negative reward causes the model to discard correct algorithms for certain training problems. Consequently, when encountering testing problems that share similar underlying concepts and require these discarded algorithms, the model remains unable to identify the correct solution, thereby degrading its performance.

To simulate RL training with false positive rewards, we blend problems with weak test cases that allow incorrect solutions to pass into our training set. As a notable example, we examined DeepCoder's RL training set. Despite their claims of providing strong test case selection criteria, we find that their tests still cannot cover all tricky edge cases - there exists incorrect solutions that can pass all their selected test cases but will subsequently fail on official test cases. After incorporating unfiltered Code-RL training data from these sources into our problem set, we observed a sudden drop on testing set accuracy midway through the training process, as shown in Figure 10. We suspect that this performance degradation occurs because the model learns to solve some training problems without considering all corner cases or by generating brute-force solutions with incorrect time complexity. This wrongly learned policy is then penalized while evaluating against strong test cases in the testing set.

### A.10 Instruction for evaluation

**Math**

```
Please reason step by step, and put your final answer within \\boxed{{}}.
```

**No Starter Code (Python)**

```
Write Python code to solve the problem. Please place the solution code in
the following format:
```python
# Your solution code here
```
```

**Has Starter Code**

```
Solve the problem starting with the provided function header.

Function header:
```
<starter_code>
```
Please place the solution code in the following format:
```python
# Your solution code here
```
```

### A.11 Example of AceReason-14B response to simple query

```
User:  hi
Model:
<think>
Alright, the user wrote "hi" in the middle of the message.

Probably just saying hello.

I should respond in a friendly and welcoming way.

Maybe ask how I can assist them today.

Keep it open-ended to encourage them to share what they need help with.
</think>
Hello!  How can I assist you today?
```

**User**: How many strawberries do I need to collect 9 r's?
**Model**:
**<think>**
First, I need to understand what the user is asking. They want to know
    how many strawberries are needed to collect 9 r's.

I interpret r's as the letter 'r'. Each strawberry is associated with a
    certain number of 'r's.

Looking at the word strawberry, I count the number of 'r's it contains.
    There are three 'r's in strawberry.

If one strawberry provides three 'r's, then to get a total of nine 'r's,
    I divide the total needed by the number each strawberry provides.

So, nine divided by three equals three.

Therefore, the user needs to collect three strawberries to have nine 'r's
    .
**</think>**

To determine how many strawberries you need to collect **9 "r's**, let's
    break it down step by step.

### Step 1: Understand the Context
- It seems you're referring to the letter **"r"** within the word **
    strawberry.**

### Step 2: Count the r's in strawberry
- Let's look at the word **strawberry**:
  - **s t r a w b e r r y**
  - **r**, **r**, **r** (there are **3 r's**)

### Step 3: Calculate the Number of Strawberries Needed
- Each **strawberry** contains **3 r's.**
- To collect **9 r's**, you would need:

  $$
  \text{Number of strawberries} = \frac{\text{Total r's needed}}{\text{r's per strawberry}} = \frac{9}{3} = 3
  $$

### Final Answer
$$
\boxed{3}
$$

