# OpenReview forum: "AceReason-Nemotron: Advancing Math and Code Reasoning through Reinforcement Learning"
_NeurIPS.cc/2025/Conference — NeurIPS 2025 poster_

### Official Review · Reviewer_iccS · 2025-06-27

**Clarity:** 4
**Significance:** 3
**Originality:** 3
**Rating:** 5
**Confidence:** 4

**Summary:**

In this paper, the authors examine the question of whether small- and mid-size reasoning models should be built by training directly with RL, or via distillation from larger and more capable reasoners. Via a relatively simple recipe involving GRPO, they train two reasoning models (which they dub AceReason models), of size 7B and 14B, and show that they can outperform comparable reasoners built via distillation. They outline the details of their training setup, their data curation strategy, and report a fine-grained analysis of how RL impacts the models' reasoning capability.

**Questions:**

My main question for the authors is to produce evidence supporting the general applicability of their training recipe. They should carry out their training pipeline on at least one model of size ~1B, and most importantly, on at least one small- or mid-sized model that has not been distilled directly from DeepSeek-R1. If the gains observed there were comparable to those observed in the main results table, that would make the paper's message much more solid and would prompt me to raise my score.

Below are more questions and remarks, preceded by the line number when they refer to a specific line in the text:
- 47: "the distilled model is first trained on math-only prompts" What distilled model is being referred to here?
- The authors make a big deal of training completely on-policy, by doing only one gradient update per rollout, but how does this happen in practice? In line 191, the authors say that their rollout batch size is 128, with group size $G=8$. Does this mean that they use a minibatch size of 128 when computing the gradient? That seems impossible. Could the authors provide details?
- 273: "Ace Reason still remains its superiority" please fix this typo.

**Ethical Concerns:**

["NO or VERY MINOR ethics concerns only"]

**Final Justification:**

The authors have clarified the main message of their work, which I believe is supported by the experiments they had show in their submitted manuscript. Moreover, they have ran an additional experiment with a smaller (1.5B) model, which further strengthens the paper's message.

**Limitations:**

The authors do not discuss the limitations of their work. In my opinion, and considering that the main message of the paper is that large-scale RL can enhance the reasoning capabilities of small- and mid-sized models, these limitations are:
- The models that authors consider are ultimately mid-sized by today's standards. Capable LLMs can have has little as 1B parameters today, but the authors do not attempt their pipeline on anything below 7B in size.
- Moreover, the authors only attempt their pipeline on two very specific models, both distilled from DeepSeek-R1, a very capable and large reasoner. This calls into question the applicability of their pipeline to generic models in that size class, and undermines the main message of the paper.

The authors should address these limitations, or at least clearly outline them in an ad-hoc paragraph.

**Paper Formatting Concerns:**

No concerns.

**Quality:**

3

**Strengths And Weaknesses:**

**Strengths**
- The question addressed by the authors is timely and pertinent.
- Their training setup is ultimately quite simple and easily reproducible.
- The gains w.r.t baseline are substantial.
- In fine-grained analyses in section 4.3 are very useful and insightful.

**Weaknesses:**
- While it is true that the DeepSeek paper had argued for distillation instead of RL for building mid-size reasoners, I do not think that the idea that a reasoner of 7B or 14B can be successfully trained via RL is as controversial or unexpected as the authors imply.
- 7B and 14B are relatively small sizes in today's day and age, but there are capable LLMs that are even smaller, e.g. Qwen2.5-1.5B. It remains to be seen whether the authors' RL pipeline would still be successful at those sizes.
- The authors only use models distilled from DeepSeek as base models in their experiments. This undermines their point on RL vs. distillation, as well as leaving the possibility that their RL pipeline would not be successful in general.

---

> ### Author Rebuttal · Authors · 2025-07-30
>
> We sincerely thank the reviewer for their thoughtful and constructive feedback. We appreciate you finding the question addressed is “**timely and pertinent**” and analysis in section 4.3 are “**very useful and insightful**”. Below, we aim to clarify and address your questions.
>
> ## Main Contribution Clarification
>
> We would like to clarify our research question and define what constitutes "successful" RL in our context.
> The DeepSeek-R1 training pipeline consists of: (1) SFT using 800K distilled long-CoT responses, followed by (2) large-scale RL. Our experiments **explicitly follow** this pipeline, starting from models already distilled from R1 (DeepSeek-R1-Distilled-Qwen-7B/14B). We intentionally do **not** experiment with RL from base models, as this represents a **different paradigm** (similar to DeepSeek-R1-Zero) that yields inferior results.
>
> For smaller models, the field has converged on two competing approaches:
> - **Distillation-only**: Generate massive amounts of distillation data (e.g., 4.9M+ examples for OpenMath-Nemotron, 700K for OpenCode-Nemotron) and conduct large-scale SFT to create specialized expert models
> - **Distillation + RL**: Start from DeepSeek-R1-Distilled checkpoints (800K SFT data) and apply large-scale RL (our approach, also used by Skywork-OR1-Math and Light-R1-14B)
>
> Our central research question is: "*For smaller models, can RL on top of distilled models achieve better performance than distillation alone?*" This question matters because recent influential work (DeepSeek-R1 and Llama-Nemotron-235B) concluded that RL yields suboptimal results for smaller models compared to distillation, recommending RL only for large models like DeepSeek-V3-671B or Llama-Nemotron-Ultra-253B.
>
> Therefore, our definition of "successful" RL is not merely improving over the initial distilled model – we aim to **match or exceed** the best specialized distillation-only expert models. As stated in Lines 43-45, “we demonstrate that large-scale RL can significantly enhance strong small- and mid-sized SFT models, achieving performance competitive with state-of-the-art distillation at 7B and surpassing it at 14B”.
>
> | Model | AIME24 | AIME25 | LiveCodeBench v5 | LiveCodeBench v6 |
> |-------|--------|--------|------------------|------------------|
> | OpenMath-14B | 76.3 | 63 | - | - |
> | OpenCode-14B | - | - | 59.4 | 51.1 |
> | AceReason-14B | **78.6** | **67.4** | **61.1** | **54.9** |
>
>
> Below, we hope to address your questions:
>
> >#1: “I do not think that the idea that a reasoner of 7B or 14B can be successfully trained via RL is as controversial or unexpected as the authors imply”
>
> We would like to clarify what we mean by "successful" RL training. Our definition of success is **not** merely achieving improvements over the initial distilled model - rather, it's matching or exceeding the best specialized distillation-only models, which is highly challenging.
> Prior work, including DeepSeek-R1 and Llama-Nemotron-235B, explicitly recommended distillation as the superior approach. The "Sober Reasoning" paper (arXiv:2504.07086 April 2025) reinforced this view, claiming that "reinforcement learning (RL) approaches yield only modest improvements - far below prior claims"
>
> In contrast, our results directly challenge this consensus. At 14B scale, AceReason surpasses the best distillation-only expert models (e.g., OpenMath, OpenCode). One of the key insights is that we reveal that model size critically determines RL effectiveness - as 7B shows competitive results and 14B shows outperforming results. Meanwhile, the distillation model OpenMath performance on math benchmarks saturated when scale model size from 14B to 32B (e.g., AIME25 drops from 63 to 62.5).
>
> Whether this finding is "controversial" may be subjective and time-dependent. To the best of our knowledge at the submission time, we believe demonstrating that RL can outperform specialized distillation at 14B scale represents novel insights to the community.
>
> >#2“1.5B…It remains to be seen whether the authors' RL pipeline would still be successful at those sizes.”
>
> Thanks for your comments. We agree 1.5B model size is also important and facilitate reproducibility of our study. Therefore, we conducted additional experiments on 1.5B models to show the effectiveness of our RL recipe. As we found below, AceReason-1.5B shows significant improvement over the initial model achieving 18.9%+ on AIME24, 12.8%+ on AIME25. We hope these results resolve your concern of the effectiveness of RL pipeline when applied to smaller model sizes.
>
> | Model | AIME24 | AIME25 | LiveCodeBench v5 |
> |-----------|------------|------------|---------------------|
> | DeepSeek-R1-Distilled-Qwen-1.5B | 28.9 | 22.7 | 16.9 |
> | AceReason-1.5B | 47.7(**+18.8**) | 35.5(**+12.8**) | 27.2(**+10.3**) |
>
>
>
> >#3“The authors only use models distilled from DeepSeek as base models in their experiments.”
>
> Thank you for your question. The choice of using models distilled from DeepSeek-R1 as the initial model in our experiment was deliberate and essential to our research design. Since our core research question examines whether RL can enhance already-distilled models following the DeepSeek-R1 pipeline (SFT → RL), we necessarily start from DeepSeek-distilled checkpoints. Using models distilled through other methods would introduce confounding variables and prevent us from isolating the effects of our RL approach. In addition, experimenting with RL from base pre-trained models (as in DeepSeek-R1-Zero) represents a fundamentally different research question and is beyond the scope of this work.
>
> >#4“The authors do not discuss the limitations of their work. … these limitations are:
> … Capable LLMs can have as little as 1B parameters today, but the authors do not attempt their pipeline on anything below 7B in size.
> the authors only attempt their pipeline on two very specific models, both distilled from DeepSeek-R1, …
> The authors should address these limitations, or at least clearly outline them in an ad-hoc paragraph.“
>
> Thanks for your comment. We put the limitation section of this work in Appendix A.1.
> To your questions of “attempt their pipeline on … 1B model” and “only attempt their pipeline on two very specific models”, we hope our previous response in #2 (add 1.5B results) and #3 (explain our experiment setting is different from DeepSeek-R1-Zero) resolve your questions.
> We appreciate your feedback on these limitations and will explicitly mention the research question clearly in the introduction, experiment setting that uses DeepSeek distilled model for RL (L113), and add 1.5B experiment results to the paper. We will also mention we are not using a pre-trained base model for RL in the limitation as it is related to a different experiment setting. Thank you!
>
> ## Technical Clarifications
> >Q1: Line 47 - "the distilled model is first trained on math-only prompts - What distilled model is being referred to here?"
>
> The "distilled model" refers specifically to DeepSeek-R1-Distilled-Qwen-7B/14B models.
>
> >Q2: On-policy training implementation details
>
> Excellent question about our implementation. Let me clarify with a concrete example:
> - With group size G=8 and batch size 128, we generate 128×8 = 1,024 examples total
> - These 1,024 examples are divided into minibatches for gradient computation
> - We accumulate gradients across all minibatches
> - Crucially: We perform exactly ONE gradient update after processing all 1,024 examples
>
> This is implemented in veRL by setting
>
> data.train_batch_size == actor_rollout_ref.actor.ppo_mini_batch_size
> . We'll add this clarification to improve clarity.
>
> We hope these clarifications and additional evidence address your key concerns while demonstrating that our work makes significant contributions to show how RL can enhance already-capable reasoning models. Please feel free to let us know if you have any other questions. Thank you!

---

> > ### Comment · Reviewer_iccS · 2025-08-04
> >
> > I thank the authors for their clarifications re: their main research question and main baselines, and even more so for the additional experiment they carried out with AceReason-1.5B. I moreover thank them for addressing also my minor concerns in their rebuttal.
> >
> > These additions definitely improve the paper and make its message much clearer and stronger. I will be raising my score to a full accept (5).

---

> > > ### Author Response · Authors · 2025-08-04
> > >
> > > Thank you so much for your thoughtful engagement throughout this review process! We're thrilled that our clarifications and the AceReason-1.5B experiment successfully addressed your concerns, and we truly appreciate you raising your score to a full accept (5). We will make sure to incorporate all these improvements into the final paper version.

---

### Official Review · Reviewer_V9Hg · 2025-07-01

**Clarity:** 3
**Significance:** 2
**Originality:** 2
**Rating:** 5
**Confidence:** 4

**Summary:**

The authors present a recipe for using large-scale reinforcement learning to enhance distillation-based reasoning models for math and coding tasks. The proposed approach specifically targets the RL phase. By carefully curating training problems and ablating nuances in the RL pipeline, the authors demonstrate that it is possible to elevate existing distilled models to approach the performance of top models on relevant benchmarks, such as AIME and LiveCodebench. The authors also incorporate a curriculum learning scheme that trains the model on math problems followed by coding tasks. This approach results in simultaneous improvements in both domains. They find that on-policy reinforcement learning is key for stable training.

**Questions:**

1. In the top right plot of Figure 2, the green line (AceReason-7B with math RL) and the red line (AceReason-7B without math RL) show a clear discrepancy in the final steps, as the latter's performance on LiveCodeBench declines. Has the authors investigated why this occurred?
2. In Section 4.3 line 305, the authors state that the difficulty score is estimated via the 7B model. It seems like this is different from the previous estimation via DeepSeek-R1, introduced in Section 3.2. I wonder why two different estimation setups are used.

**Ethical Concerns:**

["NO or VERY MINOR ethics concerns only"]

**Final Justification:**

The authors have provided detailed ablation experiments and clarifications on all weaknesses / questions I raised in my review. These mainly include my concerns about the novelty of the propose method (or pipeline) and findings from empirical experiments. I decided to raise my rating due to the above reasons.

**Limitations:**

yes

**Quality:**

3

**Strengths And Weaknesses:**

Strengths:

1. The proposed pipeline is effective and produces models that are significantly stronger than their distillation counterparts, approaching the performance of closed-source or proprietary models.
2. The proposed approach brings insights into some of the earlier unsolved "mysteries". These include the claim that distillation is often more effective for small models, the problem of entropy collapse (or explosion), and the importance of data (problem) curation.
3. The paper provides extensive ablation studies and detailed analysis on multiple aspects of the pipeline, including problem difficulty, length extension, and curriculum learning.

Weaknesses:

1. Many components in the proposed pipeline mainly involve or borrow small modification from existing work. It uses GRPO with on-policy training, length extension, and problem difficulty estimation, which are not new to the field.
2. While the experiments include ablation studies and analyses of some modifications, the justification for using each of them is still unclear to me, as most results are highly empirical. Readers are unsure whether these modifications would still be effective in a slightly different setting, such as with a different dataset or initial model.

---

> ### Author Rebuttal · Authors · 2025-07-30
>
> >“Many components in the proposed pipeline mainly involve or borrow small modification from existing work. It uses GRPO with on-policy training, length extension, and problem difficulty estimation, which are not new to the field.”
>
> Thank you for your comment. We appreciate your acknowledgement of our proposed pipeline “**approaching the performance of close-source models**”, “**brings insights into some of the earlier unsolved mysteries**” and “**provide extensive ablation studies on multiple aspects of the pipeline**”. While we agree these techniques are not new, we believe the originality of our work are as follows:
> - **Effectiveness and efficiency improvement of each technique**: our ablation demonstrates the importance of each technique used in the pipeline such as strict on-policy training (stabilize training), stage-1 8K training (improves efficiency and effectiveness compared to starting from 16K or 24K), and problem difficulty (effectiveness in later stages).
> - **Math and Code RL pipeline**: we present the first systematic demonstration of math-only RL enhances code reasoning. This cross-domain transfer is a core aspect of the insight from our study. While concurrent works (Skywork-OR1, May 28; Magistral, June 12) have mentioned this, they appeared after our submission deadline. We thus introduce a two-stage RL pipeline, combining math-only RL and code-only RL, to boost training efficiency.
> - **State-of-the-art results on both math and code**: combining all the techniques above, we are the first to show that a medium-sized SFT (distilled) model, when enhanced with RL, can outperform specialized distillation models and achieve SOTA results on both math and coding tasks, which is a non-trivial contribution and requires a significant technical depth. In contrast, many prior studies **focus exclusively** on either math or code, highlighting the difficulty of handling heterogeneous prompts and inherent complexity of RL training.
> - **Novel insights on code-RL capability improvements**: we present a series of ablation studies to understand the unique challenge of code-RL training (appendix A.8) and show a unique perspective of code-RL indeed expanding the initial model performance limit on LiveCodeBench (Figure 4 - top). In addition, we add pass@1024 results below to further validate our observation that code-RL expand capability of the initial model. We believe these insights are novel to understand the code RL training.
>
> - LiveCodeBench v6 Results:
> | pass@k | 8 | 16 | 32 | 64 | 128 | 256 | 512 | 1024 |
> |--------|----|-----| ----|----|----|-----|-----|------|
> | DeepSeek-R1-Distilled-Qwen-7B | 47.4 | 50.9 | 53.6 | 56.3 | 58.2 | 60.4 | 62.3 | 63.4 |
> | AceReason-7B | **58.1** | **61.7** | **64.8** | **67.8** | **70.0** | **71.9** | **73.3** | **74.3** |
>
>
>
> >“While the experiments include ablation studies and analyses of some modifications, the justification for using each of them is still unclear to me, as most results are highly empirical. Readers are unsure whether these modifications would still be effective in a slightly different setting, such as with a different dataset or initial model."
>
> Thank you for your comment. We agree that most results are indeed empirical as RL training itself already involves a lot of randomness and doing RL on top of strong reasoning models that generate long chain-of-thought responses is highly computational intensive. We extensively experiment with a series of techniques to combat training stability (on-policy), improve training efficiency (stage-1 8K training), curriculum learning (using hard prompts) and show the effectiveness of each component. We believe these insights are generally applicable to different models and dataset. We also propose a reliable two-stage training recipe to do math-RL and code-RL for training stability and efficiency. Each of these designs are well motivated to **improve training stability and efficiency** as a way to scale RL.
>
> In addition to 7B and 14B models, we validated the recipe on smaller model 1.5B and larger model 32B below (due to computational constraint, we only conducted math-RL for the 32B model). Our results consistently demonstrate that **our RL recipe enhances performance across all model scales**, and the conclusion that math-RL improves code generalizes effectively to the 32B model (LCB v5: 57.2 → 63.4), this is close to the DeepSeek-R1 model (65.9) even without code-RL training, which surprise us. Furthermore, to ensure the community can reproduce our study, we are committed to **open-sourcing our training dataset and model**.
>
> | Model | AIME24 | AIME25 | LiveCodeBench v5 |
> |-----------|------------|------------|---------------------|
> | DeepSeek-R1-Distilled-Qwen-1.5B | 28.9 | 22.7 | 16.9 |
> | AceReason-1.5B | 47.7(**+18.8**) | 35.5(**+12.8**) | 27.2(**+10.3**) |
> | DeepSeek-R1-Distilled-Qwen-32B | 72.6 | 54.9 | 57.2 |
> | AceReason-32B (math-only RL) | 79.0 (**+6.4**) | 70.6 (**+15.7**) | 63.4 (**+6.2**) |
>
> ## Technical questions
>
> >“In the top right plot of Figure 2, the green line (AceReason-7B with math RL) and the red line (AceReason-7B without math RL) show a clear discrepancy in the final steps, as the latter's performance on LiveCodeBench declines. Has the authors investigated why this occurred?”
>
> Our experiments reveal two distinct advantages of initializing code-RL training from a math-RL checkpoint. Firstly, we observe an improvement in code task performance. Secondly, the math-RL checkpoint consistently produces more effective chains of thought, contributing to a more stable RL training process, particularly in later stages of code-RL. The length of model responses demonstrates a gradual increase throughout training. Conversely, direct initialization from a distilled checkpoint results in fluctuating response lengths and an exponential increase in the clip-ratio (% of responses exceeding maximum length), leading to **instability** during RL training and subsequent **training collapse** in later stages. Therefore, we have elected to commence all subsequent experiments from a math-RL checkpoint.
>
> >“In Section 4.3 line 305, the authors state that the difficulty score is estimated via the 7B model. It seems like this is different from the previous estimation via DeepSeek-R1, introduced in Section 3.2. I wonder why two different estimation setups are used.”
>
> Thanks for raising this question. We would like to clarify the distinction between Section 3.2 and Section 4.3.
> In Section 3.2, we utilize DeepSeek-R1 to determine if the collected math problems are solvable by the R1 model. This process primarily serves to clean the dataset, removing question-answer pairs that may be noisy due to OCR-related errors, thus ensuring the questions are solvable at least with the R1 model.
>
> Conversely, in Section 4.3, we employ the 7B model to estimate the problem difficulty (pass rate). This approach is more appropriate as it uses the current policy model to assess the pass rate and filter out questions deemed easy by this model. These filtered data are subsequently used for training the current policy model. Intuitively, this provides a more accurate estimation of problem difficulty for the current policy model, and is beneficial for RL training.

---

> > ### Comment · Reviewer_V9Hg · 2025-08-06
> >
> > I appreciate the authors' efforts in providing additional experiments to support the method's effectiveness and in clarifying the questions I raised. I currently have no further concerns.
> >
> > I recommend including the rebuttal results in future revisions if possible. Additionally, express the novelty in both data pipelines and insights from RL training more clearly, as shown in the previous response. I will adjust the rating accordingly.

---

> > > ### Author Response · Authors · 2025-08-06
> > >
> > > We appreciate your time and effort to provide thoughtful feedback. We are glad to see our response has addressed your concerns. We will make sure to include these experiment results and express the novelty of our data pipeline and RL training insights more clearly in the final version of the paper. Thank you!

---

### Official Review · Reviewer_QBrj · 2025-07-03

**Clarity:** 3
**Significance:** 3
**Originality:** 2
**Rating:** 4
**Confidence:** 4

**Summary:**

The paper proposes a post-training method using Reinforcement Learning (RL) on small and medium-sized models. It implements staged training for math and code tasks, which improves the model's performance on these specific tasks. The paper provides a detailed account of the training setups and practical tricks used to stabilize the training process and enhance its effectiveness. The evaluation shows that RL significantly boosts the reasoning abilities of the initial model, achieving performance that is competitive with state-of-the-art reasoning models.

**Questions:**

1. Following up on W2, do you have additional experimental data with a larger pass@k to prove that RL expands the model's capability limits?
2. Does the model show any improvement in general-purpose reasoning abilities after RL training?

**Ethical Concerns:**

["NO or VERY MINOR ethics concerns only"]

**Final Justification:**

The authors provided more experiments and clarifications on the questions raised in the previous review. The responses mitigate my concerns. I decided to raise my rating.

**Limitations:**

See the weaknesses above.

**Paper Formatting Concerns:**

The paper adheres to the NeurIPS 2025 formatting guidelines without any major issues.

**Quality:**

3

**Strengths And Weaknesses:**

Strengths:
- S1: This paper is well-written and easy to follow.
- S2: The paper offers many practical training tricks, supported by experiments, for managing the instability inherent in reinforcement learning.
- S3: The paper provides a detailed analysis to demonstrate the performance improvements gained from RL.
Weaknesses:
- W1: The paper's novelty and contribution appear limited, as it seems to combine existing techniques. Examples include staged length extension and curriculum learning.
- W2: The experiments analyzing the model's capability limits are questionable. In the cited paper https://arxiv.org/abs/2504.13837 (Yue et al., 2025), the performance inflection point for the base and RL models is around pass@256. However, this paper only presents results up to pass@64, which is insufficient to prove that RL training actually expands the model's capability limits.
- W3: The paper does not evaluate the model's general reasoning abilities, for instance, on STEM or other puzzle-related reasoning tasks. Theoretically, an improved reasoning model should demonstrate stronger cross-domain reasoning skills.

---

> ### Author Rebuttal · Authors · 2025-07-30
>
> Thank you for your detailed review. We hope to address your concerns with concrete evidence:
> >“The paper's novelty and contribution appear limited, as it seems to combine existing techniques. Examples include staged length extension and curriculum learning”
>
> We agree that AceReason is built based on a combination of methods and techniques, and we appreciate your acknowledgement that our work offers “**many practical training tricks, supported by experiments**”. The novelty of our study lies in our ablation study of the necessity of stage-1 (8K) training (Fig 2b) resulted in training efficiency improvements, effectiveness of curriculum learning (Table 3) which uses harder prompts for later stage training, and demonstration of strict on-policy training which stabilizes RL training. Our analysis on problem-level accuracy also provides the first demonstration of the improvement of RL for both math and coding in Figure 4.
>
> In addition to that, we provide the systematic demonstration that math-only RL significantly improves code reasoning (6-14% on LiveCodeBench) and proposes the two-stage math-only RL and code-only RL pipeline, which stabilizes the code RL training (Figure 1b). Many prior studies focus **exclusively on either math or code**, showing the difficulty of handling heterogenous prompts and inherent complexity of RL. To the best of our knowledge, this cross-domain transfer was not explicitly mentioned in prior studies and concurrent works mentioning this (Skywork-OR1 May 28, Magistral Jun 12) appeared after our submission deadline.
>
> Combining all these techniques, we are the first to show a medium sized SFT model using RL can surpass a specialized distillation model on both math and code and achieve state-of-the-art performance at the time.  At 14B, we outperform OpenMath-14B (AIME25: 63 vs 67.4) and OpenCode-14B (LCB-v5: 59.4 vs 61.1) - the best expert models in the Qwen2.5 family. Importantly, the challenge of combining these methods to achieve significant improvements is substantial and requires non-trivial technical depth.
>
> >“Following up on W2, do you have additional experimental data with a larger pass@k to prove that RL expands the model's capability limits?”
>
> Thanks for your question. We initially used the pass@64 evaluation setting following Yue et al. [1] LiveCodeBench evaluation set up in their Figure 3. We agree showing pass@k at larger k can show more concrete results. Below, we computed pass@8 to 1024 results using the 7B model on two benchmarks (AIME2025 and LiveCodeBench v6), which are released after the release of the distilled model (DeepSeek-R1-Distilled). Each datapoint is calculated based on an average of 100 runs to reduce variance.
>
> As we can see, AceReason-7B shows improved pass@k compared to the initial distilled model across k=8 to 1024. The gap in the LiveCodeBench v6 is more than **10+ points** as the coding problems require the answer to pass all test cases, which are **difficult to guess correctly**. Our results show that RL improves pass@k on coding tasks and show evidence that RL expands capability limits.
>
> - AIME 2025 Results:
> | pass@k | 8 |  16 | 32 | 64 | 128 | 256 | 512 | 1024 |
> |--------|----|----|----|----|----|-----|-----|------|
> | DeepSeek-R1-Distilled-Qwen-7B | 62.2 | 66.8 | 71.3 | 76.5 | 81.2 | 86.2 | 90.3 | 92.6 |
> | AceReason-7B | **73.4** | **76.0** | **78.3** | **85.4** | **89.0** | **93.0** | **94.8** | **96.1** |
>
>
> - LiveCodeBench v6 Results:
> | pass@k | 8 | 16 | 32 | 64 | 128 | 256 | 512 | 1024 |
> |--------|----|-----| ----|----|----|-----|-----|------|
> | DeepSeek-R1-Distilled-Qwen-7B | 47.4 | 50.9 | 53.6 | 56.3 | 58.2 | 60.4 | 62.3 | 63.4 |
> | AceReason-7B | **58.1** | **61.7** | **64.8** | **67.8** | **70.0** | **71.9** | **73.3** | **74.3** |
>
>
> [1] Yue, Y et al., Does reinforcement learning really incentivize reasoning capacity in llms beyond the base model?
>
> > “Does the model show any improvement in general-purpose reasoning abilities after RL training?”
>
> Thanks for your comment. We agree examining the math and code RL model on other reasoning tasks can show a more broader understanding of the model’s reasoning capabilities. Below, we evaluated AceReason models on three tasks including STEM (GPQA) and logic puzzles such as (Knight & knave, Zebra-logic). Interestingly, we found AceReason shows significant improvements on logic puzzles (e.g., 12.3%+ on Knight & knave and 12.6% on Zebra-Logic for AceReason-14B). On GPQA, which is a dataset focused mainly on college-level knowledge, AceReason-14B shows 2% improvements. These results further support our previous observations that math-RL improves coding as it shows the math and code **RL improvements are generalizable** to other domains.
>
> | Model | GPQA | Knight & Knave | Zebra-Logic |
> |-------|------|----------------|-------------|
> | DeepSeek-R1-Distill-7B | 49.1 | 35.2 | 24.5 |
> | AceReason-7B (Ours) | **51.1 (+2.0)** | **56.4 (+21.2)** | **39.6 (+15.1)** |
> | DeepSeek-R1-Distill-14B | 59.1 | 83.8 | 60.1 |
> | AceReason-14B (Ours) | **61.2 (+2.1)** | **96.1 (+12.3)** | **72.7 (+12.6)** |
>
> We hope this evidence addresses your concerns and demonstrates the significance of our contributions.

---

> > ### Comment · Area_Chair_8hKm · 2025-08-05
> > **Please post your response**
> >
> > Dear Reviewer QBrj,
> >
> > This is a gentle reminder to post your response. The deadline for the author-reviewer discussion period is approaching. The authors have responded to your reviews and also to others' reviews. Please have an open discussion with the authors about your reviews and whether your concerns have been addressed.
> >
> > Best,
> >
> > AC

---

> > > ### Author Response · Authors · 2025-08-07
> > > **Follow up**
> > >
> > > Dear Reviewer QBrj,
> > >
> > > We appreciate your review and understand this is a busy period. If possible, we'd value your thoughts on our rebuttal where we provided the extended pass@k analysis and general reasoning evaluations you requested.
> > > Any feedback would help us improve our work.
> > >
> > > Thank you for your time and valuable feedback.
> > >
> > > Best regards,
> > > The Authors

---

### Official Review · Reviewer_8qff · 2025-07-05

**Clarity:** 3
**Significance:** 3
**Originality:** 3
**Rating:** 4
**Confidence:** 3

**Summary:**

This paper demonstrates that large-scale RL training can significantly enhance the math and code reasoning capabilities of mall- and mid-sized models that have undergone SFT. It proposes a staged training approach (math-only first, followed by code-only) to improve model performance. Additionally, a high-quality, challenging, and open-source dataset is constructed to support reproducible research.

**Questions:**

N/A

**Ethical Concerns:**

["NO or VERY MINOR ethics concerns only"]

**Final Justification:**

The responses mitigate some of my concerns about Weakness 1. My final Rating is higher than 4 but lower than 5.

**Limitations:**

yes

**Quality:**

3

**Strengths And Weaknesses:**

### Strengths:
- The paper introduces a novel two-stage RL training methodology(math first, then code). This not only significantly improves performance within each domain but also reveals an unexpected benefit: math-only RL substantially improves code reasoning capabilities.
- The experiments cover a variety of math and code reasoning benchmarks and compare them with multiple SOTA models, verifying the effectiveness of the method.
- This paper constructs a high-quality and challenging dataset through a rigorous pipeline, ensuring reliable verification-based RL training.


### Weaknesses:
- Although the paper indicates that RL shows a higher performance ceiling for models at 14B and above, the improvement for smaller models (e.g., 7B) is relatively limited, and it still underperforms distillation models in some math tasks.
- The paper provides a limited theoretical or empirical explanation. For example, why math RL can enhance code task performance and why code tasks (e.g., graph theory, string manipulation) benefit more.

---

> ### Author Rebuttal · Authors · 2025-07-30
>
> We sincerely appreciate your thoughtful review and recognition of our contributions on the “**novel two-stage RL training method**” and “**high-quality open-source dataset**”. We're pleased to address your concerns and provide additional results.
>
> >“Although the paper indicates that RL shows a higher performance ceiling for models at 14B and above, the improvement for smaller models (e.g., 7B) is relatively limited, and it still underperforms distillation models in some math tasks.“
>
> We appreciate this observation, which actually highlights one of our key insights: **RL effectiveness scales with model capacity**.
> Our 14B results prove the concept: AceReason-14B surpasses specialized distillation models (AIME25: 67.4% vs 63% OpenMath-14B; LiveCodeBenchv5: 61.1% vs 59.4% OpenCode-14B), demonstrating that RL can exceed state-of-the-art distillation expert models.
> For 7B models, while AceReason-7B doesn't surpass OpenMath-7B on math tasks, we achieve substantial **double-digit improvements** over the initial model. In addition, the performance on code tasks are competitive to OpenCode-7B (e.g., LCBv5 51.8% vs 51.3% OpenCode-7B).
> - AIME24: 55.5% → 69.0% (**+13.5%**)
> - AIME25: 39.0% → 53.6% (**+14.6%**)
> - LiveCodeBench v5: 37.6% → 51.8% (**+14.2%**)
>
> The performance gap with OpenMath-7B stems from fundamental training differences:
> - **Data scale**: OpenMath uses **4.9M+** math-specific samples for SFT vs. **800K** general samples in our initial model (DeepSeek-R1-Distilled-Qwen). This is at least **6x difference** in terms of math training data
> - **Specialization**: OpenMath is math-only, OpenCode is code-only; AceReason handles math, code, and general reasoning capabilities
> - **Model size scaling trajectory**: OpenMath **plateaus** from 7B→14B →32B (AIME25: 61.2→63.0→62.5), while AceReason shows strong scaling potential (7B→14B: 53.6→67.4)
>
> This scaling behavior suggests that while specialized distillation may be optimal at very small scales, RL becomes increasingly advantageous as model capacity grows at medium size.
>
> >“The paper provides a limited theoretical or empirical explanation. For example, why math RL can enhance code task performance and why code tasks (e.g., graph theory, string manipulation) benefit more.”
>
> First, we would like to provide more empirical results for the RL transfer reasoning capability phenomenon. Our investigation reveals that RL's benefits extend beyond math and coding and generalize to other tasks such as STEM (GPQA) and logic reasoning (Knight & Knave, Zebra-logic).
> | Model | GPQA | Knight & Knave | Zebra-Logic |
> |-------|------|----------------|-------------|
> | DeepSeek-R1-Distill-7B | 49.1 | 35.2 | 24.5 |
> | AceReason-7B (Ours) | **51.1 (+2.0)** | **56.4 (+21.2)** | **39.6 (+15.1)** |
> | DeepSeek-R1-Distill-14B | 59.1 | 83.8 | 60.1 |
> | AceReason-14B (Ours) | **61.2 (+2.1)** | **96.1 (+12.3)** | **72.7 (+12.6)** |
>
> In addition, our detailed analysis (Figure 8 and Section A.7) reveals how these enhanced reasoning capabilities specifically benefit code generation:
> - **Improvements in algorithmic tasks**: Code problems requiring mathematical thinking (combinatorics, counting, number theory) show the largest gains, with direct application of math reasoning.
> - **Gains in logic-heavy tasks**: Problems involving complex control flow, constraint handling, and systematic search (graph algorithms, dynamic programming) benefit from enhanced reasoning capabilities.
> - **Improvements in implementation-focused tasks**: Even string manipulation and data structure problems show gains, suggesting that better reasoning helps with problem decomposition and solution planning.
>
> We emphasize that our contribution is primarily empirical - we document and quantify these transfer effects systematically across multiple domains. Providing definitive theoretical explanations for these emergent behaviors remains beyond the scope of this work and represents an important direction for future research. We will make this clear in the limitation section of this paper. Thank you!

---

> > ### Comment · Reviewer_8qff · 2025-08-05
> >
> > Thanks for the detailed responses. The explanations mitigate some of my concerns about Weakness 1. But I still believe my ratings are consistent with the contribution of this work.

---

> ### Author Response · Authors · 2025-08-06
>
> Dear Reviewer 8qff,
>
> Thank you for taking the time to engage with our rebuttal and for acknowledging that our explanations have mitigated some of your concerns about Weakness 1. We genuinely appreciate your thorough review and constructive feedback.
>
> Thank you again for your time and insights.
>
> Best regards,
> The Authors

---

### Decision · Program_Chairs · 2025-09-17

**Decision:**

Accept (poster)

**Comment:**

The paper investigates whether large-scale RL can enhance the reasoning capabilities of small- to mid-sized models beyond distillation-based approaches. The authors propose a two-stage RL training pipeline (math-first, then code) and demonstrate significant improvements on math and code benchmarks. Key findings include: (1) math-only RL improves both math and code performance, with extended code-only RL further boosting code results, (2) curriculum learning with progressive response lengths stabilizes training, and (3) on-policy RL is critical for performance. The work also releases a high-quality, open-source dataset for verification-based RL.

Reviewers acknowledge the novel two-stage RL methodology that improves performance in both domains and reveals math RL’s unexpected benefit for code reasoning. The rigorous data curation pipeline and extensive ablations provide actionable insights for reproducible research. The practical training tricks address RL instability and outperform distillation baselines. The gains are substantial, with the 14B model approaching proprietary model performance. A high-quality open dataset enabling reproducible RL training is also highlighted.

On the other hand, reviewers note limitations. The 7B model’s improvement is modest, and it underperforms distillation in some math tasks. Theoretical explanations for cross-domain benefits are lacking. Generalization to smaller models or non-DeepSeek base models remains unverified.

In general, the paper makes valuable contributions by demonstrating RL’s effectiveness in enhancing reasoning capabilities of small- and mid-sized models, supported by solid experiments, a useful dataset, and insightful analyses. While it has limitations, its strengths such as novel methodology, significant performance gains, open dataset, and reproducibility outweigh the weaknesses.

During the author rebuttal, the authors have clarified the main message of their work, and have run additional experiments to address reviewers' concerns. These efforts are appreciated by most reviewers.